# Flexible High-Performance and Screen-Printed Symmetric Supercapacitor Using Hierarchical Rodlike V_3_O_7_ Inks

**DOI:** 10.3390/nano13162282

**Published:** 2023-08-08

**Authors:** Baoying Lin, Yinyin Zheng, Jinglu Wang, Qian Tu, Wentao Tang, Liangzhe Chen

**Affiliations:** School of Electronic Information Engineering, Jingchu University of Technology, Jingmen 448000, China; lby15160232827@126.com (B.L.); z13299251453yy@163.com (Y.Z.); w17720520749@126.com (J.W.); tq010406@126.com (Q.T.)

**Keywords:** V_3_O_7_ nanorod, ink, screen printing, supercapacitor, energy storage device

## Abstract

The emergence of the Internet of things stimulates the pursuit of flexible and miniaturized supercapacitors. As an advanced technology, screen printing displays vigor and tremendous potential in fabricating supercapacitors, but the adoption of high-performance ink is a great challenge. Here, hierarchical V_3_O_7_ with rodlike texture was prepared via a facile template–solvothermal route; and the morphology, component, and valence bond information are characterized meticulously. Then, the screen-printed inks composed of V_3_O_7_, acetylene black, and PVDF are formulated, and the rheological behaviors are studied detailedly. Benefitting from the orderly aligned ink, the optimal screen-printed electrode can exhibit an excellent specific capacitance of 274.5 F/g at 0.3 A/g and capacitance retention of 81.9% after 5000 cycles. In addition, a flexible V_3_O_7_ symmetrical supercapacitor (SSC) is screen-printed and assembled on the Ag current collector, exhibiting a decent areal specific capacitance of 322.5 mF/cm^2^ at 0.5 mA/cm^2^, outstanding cycling stability of 90.8% even after 5000 cycles, satisfactory maximum energy density of 129.45 μWh/cm^2^ at a power density of 0.42 mW/cm^2^, and remarkable flexibility and durability. Furthermore, a single SSC enables the showing of an actual voltage of 1.70 V after charging, and no obvious self-discharge phenomenon is found, revealing the great applied value in supply power. Therefore, this work provides a facile and low-cost reference of screen-printed ink for large-scale fabrication of flexible supercapacitors.

## 1. Introduction

The emergence of smartphones, tablets, wearables, IoT devices, etc., stimulate the pursuit of miniaturized and flexible energy storage devices, such as a solar cell, an ions battery, a supercapacitor, and so on [1,2,3,4]. As an emerging energy storage device, the supercapacitor has drawn much attention recently, which has a superior energy density and power density than that of lithium-ion batteries and capacitors [5,6,7,8]. Therefore, the development of the flexible supercapacitor is of great significance. To date, cumbersome fabrication technologies (e.g., electrodeposition, laser etching, electrospinning, physical or chemical evaporation) [9,10,11,12] still take the lead, while the traditional mechanical methods are incapable of maintaining the consistency of electrodes [13,14]. Hence, there is an urgent need to seek compatible fabrication technology for supercapacitors.

Screen printing technique with the merits of environmental friendliness, full adaptability, and scaling with high throughput, has shown vigor and tremendous potentiality in flexible electronic devices [15,16]. Although numerous materials emerged in early reports [17,18], the fabrication of high-performance ink is yet a great challenge. As a pseudocapacitive material, vanadium oxides (e.g., V_3_O_7_, V_2_O_5_, VO_2_, and V_2_O_3_) store/release charges by adsorption/desorption or/and intercalation/de-intercalation processes followed by redox reactions [19], resulting in an extremely fast charge/discharge rate, which enables storing and releasing more energy [20,21]. Among these, V_3_O_7_ with mixed valence states is considered to be a promising electrode material on account of the outstanding theoretical specific capacitance [22]. In addition, the unique physicochemical construction generates the ability to act as a positive and negative material synchronously, which can effectively solve the problem of charge non-conservation. Gao et al. synthesize the V_3_O_7_·H_2_O nanoribbons by a solvothermal method, and the prepared electrode exhibits a specific capacitance of 409 mAh/g at 10 mA/g [23]. More recently, the rod-like structure of the nanomaterial has proved to be an excellent architecture for electrochemical reactions. Taking the advantages of the solvothermal route (e.g., facile, large-scale, and low-cost) [24], it is promising to formulate V_3_O_7_ nanorods into inks for screen-printed electrodes.

Herein, we present a high-performance ink based on the V_3_O_7_ with a rodlike nanostructure for the flexible supercapacitor. The morphology, component, and valence bond information of V_3_O_7_ nanorods are characterized thoroughly and the possible formation mechanism is raised. Then, the screen-printed inks composed of V_3_O_7_, acetylene black, PVDF, and NMP are formulated and the rheological behaviours are explored in detail. Profiting from the splendid rheology, various patterns are printed graphically on different substrates. Importantly, the electrochemical analysis reveals better capacitive performances and stability of the printed electrode with 80-meshes, which can be attributed to the orderly aligned ink by screen printing. Moreover, a symmetrical all-solid-state V_3_O_7_ supercapacitor with excellent durability is screen-printed and assembled on an Ag layer, which can represent a remarkable energy density and cycling stability. The utilization of V_3_O_7_ ink offers a feasible and low-cost option for flexible energy storage devices.

## 2. Materials and Methods

### 2.1. Materials and Chemicals

Vanadium pentaoxide (V_2_O_5_), ethanol (EtOH), sodium sulfate (Na_2_SO_4_), N-methyl-2-pyrrolidone (NMP), and polyvinyl alcohol (PVA) were purchased from Shanghai Maclin Biochemical Technology Co., Ltd. (Shanghai, China). Ag slurry and polyethene terephthalate (PET, 0.2 mm in thickness) film were traded commercially. Nickel foam (1.0 mm in thickness) and polyvinylidene fluoride (PVDF) emulsion were bought from Saibo Electrochemical Materials Co., Ltd. (Beijing, China). All reagents were of analytical purity without treating further. The vanadium heptoxide monohydrate (V_3_O_7_·H_2_O) template was prepared referring to the previous study [25]. Deionized water (H_2_O) was used during the whole experiment.

### 2.2. Preparation of V_3_O_7_ Nanorod

V_3_O_7_ nanorod was synthesized via a facile template–solvothermal method. In short, 0.9 g V_2_O_5_, 0.05 g V_3_O_7_·H_2_O template (the preparation procedure can be found in Appendix A), 10 mL EtOH, and 30 mL H_2_O were mixed ultrasonically and transferred into a Teflon container, followed by heating at 180 °C for 12 h. After cooling to room temperature, the residue was taken out by centrifugation and washed with EtOH and H_2_O alternately. Finally, the black–green V_3_O_7_ nanorod was collected by a freeze-drying method.

### 2.3. Formulation of V_3_O_7_ Ink

The screen-printed inks were made up of V_3_O_7_, acetylene black, and PVDF emulsion (the mass ratio is 8:1:1). Initially, 80 wt% V_3_O_7_ and 10 wt% acetylene black were dissolved in NMP solution with unceasing stirring. After that, a 10 wt% PVDF emulsion was added to form a homogeneous ink. It is worth noting that the viscosity can be adjusted by the additive NMP solution.

### 2.4. Screen Printing of Electrode and Symmetric Supercapacitor

As for the electrodes, the formulated inks were squeezed out of the silk screen by a scraper and deposited on the foamed nickel. Then, the screen-printed electrodes were acquired after drying in a vacuum oven at 80 °C overnight to remove the residual solvent. The total loading mass is kept from 3.0 to 4.0 mg. According to the mesh of silk stencil (60, 80, and 100), the obtained electrodes are marked as SP-60, SP-80, and SP-100, respectively.

The procedure of supercapacitors can be summarized in three steps. In step one, the Ag slurry was screen-printed on a PET substrate to form an interfinger pattern (the effective area for printed ink is 4.38 cm^2^), followed by drying in a vacuum for 120 °C. In step two, the prepared V_3_O_7_ ink was screen-printed on the Ag layer together with drying in a vacuum for 80 °C. In step three, the prepared PVA/Na_2_SO_4_ gel (4.0 g PVA and 1.4 g Na_2_SO_4_ were dissolved in 40 mL H_2_O with unceasing stirring at 90 °C to become transparent and clear, and then the mixture was rested in the air at 25 °C overnight) is evenly covered on it, and the flexible all-solid-state V_3_O_7_ SSC is obtained after naturally drying in the air overnight.

### 2.5. Electrochemical Measurements

The electrochemical performances of electrodes in this trial are evaluated on a CorrTest electrochemical workstation (CS350H, Wuhan Corrtest Instruments Co., Ltd., Wuhan, China) with a three-electrode system in a 1.0 mol/L Na_2_SO_4_ solution at room temperature. In the three-electrode system, the screen-printed electrode served as the working electrode and was dipped ~1.0 cm into the electrolyte; and an Ag/AgCl electrode and Pt plate were used as the reference and counter electrode, respectively. The specific capacitance (*C*, F/g) could be counted according to the galvanostatic charge/discharge (GCD) curves based on the following equation [26]:(1)$$C=\frac{I\times \Delta t}{m\times \Delta V}$$
where *I*, Δ*t*, *m*, and Δ*V* are the current density, the discharge time, the mass loading of active materials, and potential windows, respectively.

The electrochemical performances of supercapacitors in this experiment are explored with a two-electrode system at room temperature. For charge balance, the mass loading of active materials in positive (*m*^+^) and negative (*m*^−^) electrodes should follow the following formula [27]:(2)$$\frac{m^{+}}{m^{-}}=\frac{C^{-}\times \Delta V^{-}}{C^{+}\times \Delta V^{+}}$$

As such, electrodes with a similar loading mass were adopted as the positive and negative electrodes to fabricate the symmetric supercapacitor.

### 2.6. Materials Characterizations

The microstructure was characterized by a field emission scanning electron microscope (SEM, Zeiss SIGMA, Oberkochen, Germany) under an accelerating voltage of 15.0 kV and a transmission electron microscope (TEM, FEI Tecnai G2 F20, Hillsboro, OR, USA) under an accelerating voltage of 30.0 kV. The component and valence bonds were measured by an X-ray photoelectron spectroscopy (XPS, ThermoFisher EscaLab250Xi, Waltham, MA, USA) using Al Kα radiation. The crystal structure was carried out by X-ray diffraction (XRD, Rigaku Mini Flex600, Tokyo, Japan) with Cu Kα radiation (λ = 1.5406 Å). The specific surface area and pore-size distribution were recorded on a specific surface area analyzer (Micromeritics ASAP 2460, FL, USA) at 77 K. The rheological behaviour of inks was performed on a rotational rheometer (Malvern Kinexus Pro+, Malvern, UK).

## 3. Results

### 3.1. Microscopic and Structural Analysis

The morphology of the samples is characterized by SEM. As a raw material, the V_2_O_5_ nanomaterial presents an irregular granular structure (Appendix A), while the V_3_O_7_·H_2_O template displays a rod construction (Appendix A). With the aid of the template, the regular and slender V_3_O_7_ nanorod emerges as created in Figure 1a and Appendix A. Compared with the template, the prepared nanorods are smaller in length. Note that this rodlike architecture is stable and enables a higher specific area for ionic migration [28]. The transmission electronic microscope (TEM) image in Figure 1b confirms the rod-like structure of the V_3_O_7_ with a width of ~124 nm. The high-resolution TEM (HR-TEM) image in Figure 1c reveals the clear lattice fringe and the d-spacing is measured as ~0.35 nm, corresponding to the (320) plane of V_3_O_7_ nanorod. In addition, the selected area electronic diffraction (SAED) pattern in Figure 1d with the distinct transmission spots suggests the monocrystal nature of the V_3_O_7_ nanorod, and the transmission spots belonging to the (310) and (320) lattice planes are observed, which is in line with the XRD results. Moreover, the elemental mapping images in Figure 1e suggest that the distribution of V and O elements is uniform on the surface of the nanorod, and a small amount of C elements may be attributed to the residual ethanol. The crystalline feature of V_3_O_7_ nanorod is explored by XRD and the result is shown in Figure 1f. In the V_3_O_7_ pattern, the major peaks are located at 2θ = 10.6°, 10.9°, 18.5°, 25.4°, 26.4°, and 32.8°, which are well indexed as the (200), (110), (310), (320), (011), and (520) planes of the orthorhombic V_3_O_7_·H_2_O (space group: Pnam, JCPDS 85-2401) with lattice constants a = 16.93 Å, b = 9.36 Å, and c = 3.64 Å [29], indicating the successful preparation of the V_3_O_7_ nanorod. In addition, the characteristic peaks assigned to the pristine V_2_O_5_ cannot be observed in the V_3_O_7_ pattern, implying high purity.

The ultimate compositions and valence information of the sample is carried out by XPS. As displayed in Figure 1g, O and V, as well as C elements, exist in the survey spectrum, which is in line with the result of elemental mapping. As exhibited in Figure 1h, the V 2p spectrum is associated with two spin-orbit splitting peaks of V 2p_3/2_ and V 2p_1/2_, which can be further divided into four peaks; i.e., the binding energies at 516.1 and 524.0 eV conform to V^4+^, while those at 517.5 and 525.5 eV correspond to V^5+^ [30,31]. The co-existence of two valence states can be assigned to the partial reduction from V^5+^ to V^4+^ by the additional ethanol. In addition, in the high-resolution spectrum of O 1s, the two peaks around 530.2 and 531.6 eV are assigned to the components of vanadium–oxygen (V-O) and hydrogen–oxygen (H-O) bonds, respectively [32]. To sum up, the feasible formation mechanism of V_3_O_7_ is as follows:$$V_{2}O_{5}\overset{xH_{2}O}{\to }V_{2}O_{5}\cdot xH_{2}O\overset{\mathrm{ethanol}}{\to }V_{3}O_{7}\cdot H_{2}O$$

In mixed solution, V_2_O_5_ nanoparticles are combined with several H_2_O molecules to form the V_2_O_5_·*x*H_2_O. Then, this hydrating compound is partially reverted by the reductive ethanol, and the V_3_O_7_·H_2_O nanorod comes into being. Furthermore, the N_2_ adsorption/desorption isotherm of V_3_O_7_ is performed to investigate the specific surface area and pore-diameter distribution. As indicated in Figure 1i, the V_3_O_7_ sample exhibits a type-IV isotherm with an H3 hysteresis loop, manifesting a typical mesoporous structure. The Brunauer–Emmett–Teller (BET) surface area is calculated as 101.2 m^2^/g with a total pore volume of 0.187 cm^3^/g. Additionally, it is found in the illustration of Figure 1i that the pore size of V_3_O_7_ primarily centers at 2.3 nm, which is fit for the supercapacitor within the range of 2.0–5.0 nm [33,34]. According to the Barret–Joyner–Halenda (BJH) model, the average pore diameter is counted as 8.0 nm. More importantly, abundant mesopores enable more active sites for ion and electron migration, thus boosting the electrochemical performances of electrode materials.

### 3.2. Ink Performances

To investigate the feasibility of screen printing, the V_3_O_7_ ink is formulated as depicted in Figure 2a. The black V_3_O_7_ ink with a sticky state exhibits outstanding fluidity in an inclined state, suggesting its suitability in the screen-printing process. Then, various patterns, including a rose, a cat, flower petals, Chinese characters, and English letters of “Jingchu University of Technology”, are easily screen-printed on paper. Moreover, it is evident that the V_3_O_7_ inks can be extruded from the meshes onto different substrates, e.g., paper, cotton cloth, and foam nickel; and these QR codes can be facilely recognized by mobile phones, confirming the applicability of the resulting ink. The rheological behaviors account for the above practicability. As revealed in Figure 2b, a typical shear-thinning appeared with the growing shear rate, manifesting that V_3_O_7_ ink has non-Newtonian fluid properties [35], which can bear the continuous extrusion during screen printing. When the shear rate increases from 0.01 to 1000 s^−1^, the viscosity declines from 4426.0 to 0.2 Pa·s. Figure 2c shows the viscosity evolution as a function of low (0.1 s^−1^) and high (100 s^−1^) shear rates. To begin with, the viscosity has a high level when the shear rate is 0.1 s^−1^, and a sharp drop in viscosity can be found when the shear rate grows to 100 s^−1^. Then, instant recovery occurs when the shear rate is restored. Impressively, V_3_O_7_ ink can restore the initial level even at a high shear rate, indicating the superior elastic rheological property. To further test and verify the application of V_3_O_7_ ink, screen stencil with different meshes (e.g., 60, 80, and 100 meshes) and nickel foam substrate are utilized to print electrodes; the optical photographs of screen-printed electrodes are presented in Figure 2d–f and Appendix A. Apparently, all screen-printed electrodes display an orderly and denser surface, resulting from the appropriate shear-extrusion process by screen printing, as can be seen in Figure 2g. It is emphasized that the ionic transport in the electrolyte can be facilitated by this surface construction [36]. Compared with SP-60 and SP-100 electrodes, the SP-80 electrode possesses a more homogeneous and complete ink layer (Appendix A and Figure 2e). The possible reason is that oversized or undersized meshes will be the obstacle in the ink transfer, and the stencil with 80-meshes is a favorite in the screen-printing process, which enables the consecutive extrusion of V_3_O_7_ ink. Moreover, the thickness of the ink layer and blank Ni foam is measured as 0.51 and 0.37 mm, respectively (Figure 2f).

### 3.3. Electrochemical Study

Figure 3 indicates the electrochemical performances of the as-prepared V_3_O_7_ electrodes. In CV curves (Figure 3a), the SP-80 electrode owns a greater integral area than that of SP-60 and SP-100 electrodes, showing superior capacitive performance. This viewpoint is held up by the GCD test, in which the SP-80 has a maximum discharge time (Appendix A). Figure 3b shows the CV curves at different scan rates of the SP-80 electrode. The quasi-rectangular shape in all curves suggests the ideal capacitive behaviour and outstanding reversibility of V_3_O_7_ materials. Also, the GCD curves of different electrodes at 0.3–5 A/g current densities are provided in Figure 3c and Appendix A. It is clear that the charge and discharge time of all GCD curves is approximately equal, making clear the high Coulombic efficiency [37]. According to Equation (1), the calculations of the specific capacitance are created in Figure 3d. With the increasing scan rates, the specific capacitance of all electrodes decreases. When at a current density of 0.3 A/g, the specific capacitance of SP-80 is ~274.5 F/g, which is greater than that of SP-60 (245.5 F/g) and SP-100 electrodes (236.0 F/g). Such a conclusion is caught even at high current density, demonstrating the best capacitance performance among all electrodes, which may impute the homogeneous and complete surface texture of the SP-80 electrode.

Moreover, the cycling stability of all samples is investigated by successive charge and discharge for 5000 cycles at a current density of 5 A/g. In Figure 3e, SP-80 retain a capacitance retention rate of 81.9%, which is higher than that of SP-60 (76.5%) and SP-100 electrodes (65.5%), demonstrating outstanding electrochemical stability for three samples. Further, electrochemical impedance spectroscopy (EIS) is adopted to explore the electrochemical reaction kinetic. All the EIS tests are measured before cycling. It is distinguishable that all Nyquist plots consist of a quasi-semicircle at the high-frequency region and a straight line at the low-frequency region. The equivalent circuit (inset of Figure 3f) is applied to fit the Nyquist plots, where *R_s_* and *R_ct_* represent the electrolyte resistance and charge-transfer resistance, and *CPE* and *W* refer to the constant phase angle element and Warburg resistance. As indicated in Figure 3f and Appendix A, all the Nyquist plots are fitting well with an error rate of less than 15%. Among these, the *R_ct_* value of SP-80 (0.10 Ω) is much smaller than that of SP-60 (0.30 Ω) and SP-100 (0.25 Ω) electrodes, manifesting faster ion diffusion and better electrochemical performances. Taken together, the SP-80 electrode is more popular than others for supercapacitors.

### 3.4. Characterization of the V_3_O_7_ SSC

To investigate the practicability of the rod-like V_3_O_7_, the all-solid-state and all-printed symmetric supercapacitor is assembled, and the preparation procedure is diagramed in Figure 4. Initially, the Ag slurry is screen-printed on a PET substrate to form an interfinger pattern, followed by drying in a vacuum for 120 °C. Then, the prepared V_3_O_7_ ink is screen-printed on the Ag-based pattern together with the drying in a vacuum for 80 °C. Finally, the PVA/Na_2_SO_4_ gel is evenly covered on it, and the flexible all-solid-state V_3_O_7_ SSC is obtained after drying naturally in the air overnight. The corresponding 80-meshes plate, interfinger pattern, and supercapacitor are shown in Appendix A.

In order to verify the optimal potential window, CV curves in different potential ranges at a scan rate of 30 mV/s are carried out. As shown in Figure 5a, when the potential window is over 1.70 V, a distinct polarization with a rapid growth in current density is discovered at a high potential position. Hence, the ideal operation potential of the V_3_O_7_ SSC can afford up to 1.70 V. Figure 5b presents the CV curves at different scan rates in a potential of 1.70 V. It is evident that the CV curves with a distorted nearly-rectangular shape can retain stable without distinct polarization even at high scan rate, indicating the low internal resistance and excellent charge/discharge capability. Figure 5c displays the GCD curves at different current densities from 0.5 to 3.0 mA/cm^2^; the discharging time is almost identical with the charging ones in all curves, suggesting the high Coulombic efficiency and good capacitance matching. The areal-specific capacitance (*C_s_*), energy density (*E*), and power density (*P*) are acquired using the following equations:(3)$$C_{S}=\frac{I\times \Delta t}{S\times \Delta V}$$
(4)$$E=\frac{1}{2}C_{S}\times \Delta V^{2}$$
(5)$$P=\frac{E}{\Delta t}$$
where *S* and Δ*V* are the effective printing area and the operating window of the SSC.

According to the GCD curves, the calculations of the *C_s_* are described in Figure 5d. Remarkably, the *C_s_* value of the V_3_O_7_ SSC is as high as 322.5 mF/cm^2^ at 0.5 mA/cm^2^, indicating a good capacitance characteristic at low current density. To expose the relationship between energy density and power density, the Ragone plots are diagramed in Figure 5e and Appendix A. The V_3_O_7_ SSC can yield a maximum energy density of 0.13 mWh/cm^2^ at a power density of 0.42 mW/cm^2^, and retain a maximum power density of 2.55 mW/cm^2^ at an energy density of 0.02 mWh/cm^2^, which precedes those of previous research about vanadium oxide-based supercapacitors, such as α-V_2_O_5_ SSC (0.48 μWh/cm^2^ at 0.11 mW/cm^2^) [38], V_2_O_5_/FTO SSC (7.70 μWh/cm^2^ at 0.36 mW/cm^2^) [39], double-layer VO_2_ SSC (0.80 μWh/cm^2^ at 0.02 mW/cm^2^) [40], V_2_O_5_·H_2_O/graphene SSC (1.13 μWh/cm^2^ at 0.01 mW/cm^2^) [41], V_2_O_5_@PEDOT/graphene SSC (0.18 μWh/cm^2^ at 0.01 mW/cm^2^) [42], and MXene-TiS_2_//MWCNTs-VO_2_ ASC (32.50 μWh/cm^2^ at 1.20 mW/cm^2^) [43]. Additionally, the cycling property is evaluated at a current density of 3.0 mA/cm^2^. As depicted in Figure 5f, the V_3_O_7_ SSC show a prominent cycling stability of 9.2% loss even after 5000 cycles. In the illustration, the post-charged V_3_O_7_ SSC shows an actual voltage of 1.70 V detected by a multimeter, and no obvious self-discharge phenomenon is found, revealing the great applied value in supply power. Furthermore, the mechanical performances of the prepared devices were investigated thoroughly. In the flexibility trials, the SSC is conducted by bending at different angles and recovering, while that is operated by bending 180° many times in the flexibility trials, and the corresponding CV curves are produced in Figure 5g,h, respectively. All curves overlapped well even bending 180° for several times. The areal-specific capacitance (*C_S_*) is calculated from the CV curves by the following equation:(6)$$C_{S}=\frac{\int I\left(V\right)dV}{S\times v\times \Delta V}$$
where ∫*I*(*V*)*dV* is the integral area, *S* is the total screen-printed area of SSC, and *v* is the scan rate. The calculations according to Equation (6) are demonstrated in Figure 5i. It is satisfactory that the *C_s_* values slightly fluctuated as a function of the bending angle (from 26.2 to 28.2 mF/cm^2^) and bending times (from 29.2 to 35.2 mF/cm^2^), representing splendid flexibility and durability. Therefore, it is concluded that the V_3_O_7_ ink with remarkable electrochemical performances for flexible all-solid-state supercapacitors holds up the potential applications in smart wearable devices.

## 4. Conclusions

In summary, a hierarchical V_3_O_7_ nanorod is prepared through a facile template–solvothermal route, which shows a BET surface area of 101.2 m^2^/g with an average pore diameter of 8.0 nm. The resulting V_3_O_7_ inks own excellent rheological behaviour and applicability. By combining with screen printing, the optimal screen-printed electrodes with orderly aligned inks exhibit an excellent specific capacitance (274.5 F/g at 0.3 A/g) and cycling stability (81.9% after 5000 cycles). In addition, the flexible and all-solid-state V_3_O_7_ SSC displays can yield a maximum energy density of 0.13 mWh/cm^2^ at a power density of 0.42 mW/cm^2^, as well as the advantage of remarkable flexibility and durability. Moreover, a single V_3_O_7_ SSC after charging enables the performance of an actual voltage of 1.70 V without an obvious self-discharge phenomenon, which can offer a new strategy for the large-scale fabrication of flexible supercapacitors in smart wearables.

## Figures and Tables

**Figure 1 nanomaterials-13-02282-f001:**
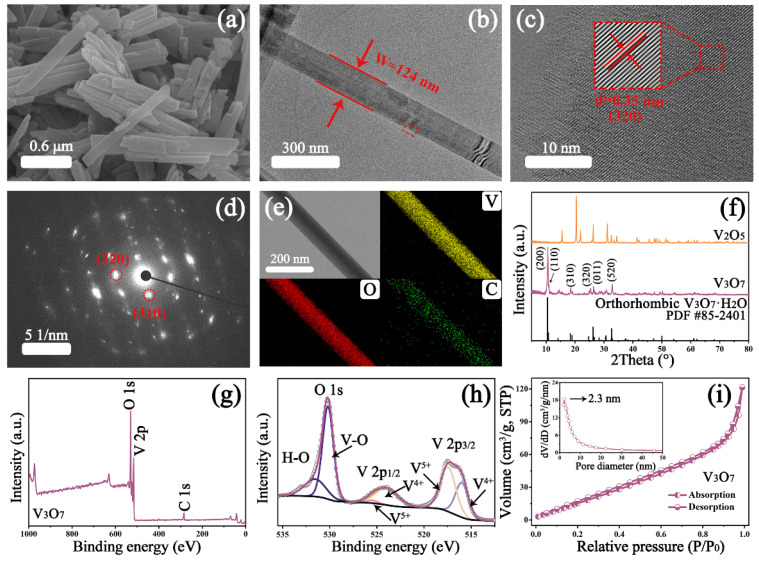
(**a**) SEM image, (**b**) TEM image, (**c**) HR-TEM image, (**d**) SAED pattern, and (**e**) elemental mapping of V_3_O_7_ nanorods; (**f**) XRD patterns of V_2_O_5_, V_3_O_7_, and the standard card for comparison; (**g**) XPS survey spectrum of V_3_O_7_ and (**h**) high-resolution spectrum of V 2p and O 1s; (**i**) the N_2_ adsorption/desorption isotherm of V_3_O_7_ sample; inset shows the corresponding distribution of pore size.

**Figure 2 nanomaterials-13-02282-f002:**
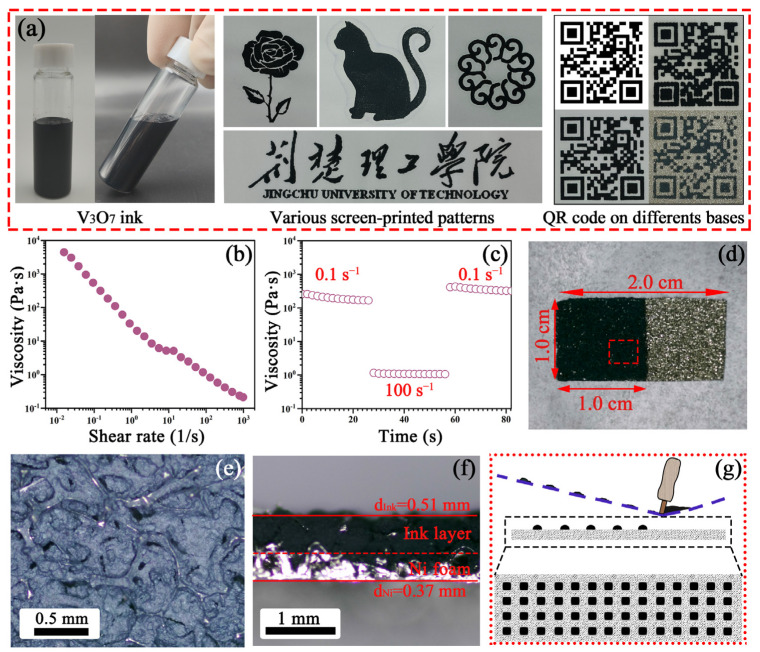
(**a**) Optical images of the V_3_O_7_ ink in the normal and slant states, various screen-printed patterns, and QR codes on different substrates; (**b**) viscosity of the V_3_O_7_ inks versus shear rate; (**c**) viscosity evolution of the V_3_O_7_ ink as a function of low (0.1 s^−1^) and high (100 s^−1^) shear rate; (**d**) the top view; (**e**) high-magnification and (**f**) cross-sectional images of the SP-80 electrode; (**g**) schematic illustration of the orderly aligned inks via screen printing.

**Figure 3 nanomaterials-13-02282-f003:**
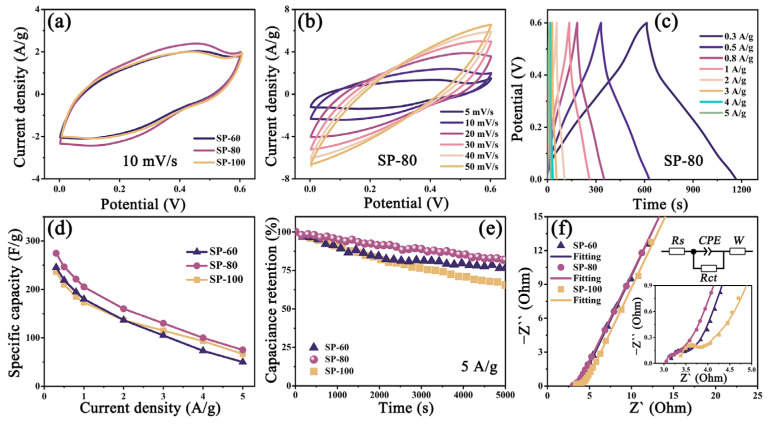
(**a**) CV curves at a scan rate of 10 mV/s of SP-60, SP-80, and SP-100 electrodes; (**b**) CV curves at different scan rates and (**c**) GCD curves at different current densities of SP-80 electrode; (**d**) specific capacitance at various current densities; (**e**) cycling stability at a current density of 5 A/g and (**f**) the Nyquist plots of SP-60, SP-80, and SP-100 electrodes; insets show the equivalent circuit and the Nyquist plots at the high-frequency region, respectively.

**Figure 4 nanomaterials-13-02282-f004:**
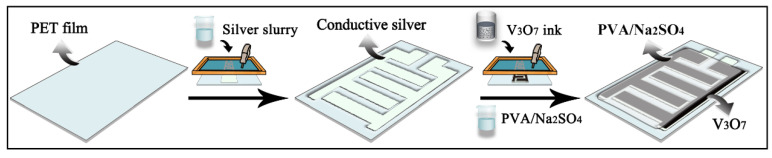
Schematic illustration of the preparation procedure of the flexible all-solid-state V_3_O_7_ symmetric supercapacitor.

**Figure 5 nanomaterials-13-02282-f005:**
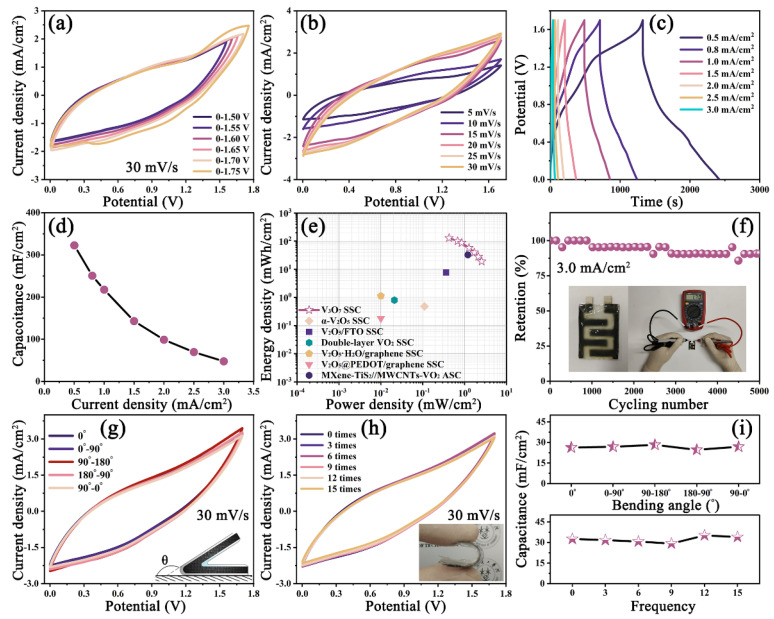
(**a**) CV curves of the V_3_O_7_ SSC at 30 mV/s under different potential windows, (**b**) CV curves at different scan rates, (**c**) GCD curves at different current densities, (**d**) the corresponding specific capacitance, (**e**) the Ragone plots, (**f**) the cycling stability at 3.0 mA/cm^2^, CV curves acquired at 30 mV/s at different (**g**) bending angle and (**h**) bending frequency, and (**i**) the corresponding areal-specific capacitance.

## Data Availability

Not applicable.

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
