# Peer review of "Flexible High-Performance and Screen-Printed Symmetric Supercapacitor Using Hierarchical Rodlike V3O7 Inks"

_nanomaterials, 2023, doi:10.3390/nano13162282_

Round 1
Reviewer 1 Report
The paper "Flexible high-performance and screen-printed symmetric supercapacitor using hierarchical rodlike V3O7 inks" is a study of V3O7-based electrode material aimed at investigating the possibility of its use in the formation of modern flexible supercapacitors. The work will be of interest to Nanomaterials readers involved in the preparation of hierarchically organised materials based on transition metal oxides and printing of planar nanostructures on their basis, however, for its publication the authors need to address a number of comments and questions:
1. In the introduction the authors need to clarify the information concerning the mechanism of charge storage, characteristic for vanadium oxides. It is stated (lines 42-43) that they accumulate energy according to the electric double layer principle, while there are works (https://doi.org/10.1038/s41467-018-03700-3; https://doi.org/10.1038/s41598-022-25707-z), stating that these materials can also demonstrate pseudocapacitive behavior.
2. It is necessary to specify how the V3O7 used as template in the process of solvothermal synthesis was obtained. If it was purchased, as well as V2O5, it is necessary to specify the manufacturer's company in Section 2.1. In addition, a SEM image of the V3O7-template should be provided as was done for the original V2O5.
3. What is the viscosity and surface tension of the functional ink used when printing the individual electrodes on the nickel foam surface and the ink used to form the symmetric supercapacitor?
4. It should be specified what amount of NMP was used for the preparation of functional inks. In addition, it is necessary to specify how the residues of this solvent were removed from the formed coatings after they were printed, since it is a high boiling solvent (boiling point of about 200 °C). Or it is necessary to clarify how important is the presence of this solvent in the composition of electrodes in the context of its influence on their electrochemical characteristics.
5. Sections 2.5. and 2.6. should be reversed in order to follow the chronological order of the research, since the electrochemical studies, as far as it follows from the article, were the final stage of the work.
6. The active material mass applied to the nickel foam surface needs to be specified. Was it the same for all three samples (SP-60, SP-80 and SP-100)? It would also be good to indicate the area of the electrodes obtained after printing the final device.
7. What is the reason for the choice of substrate (nickel foam) when studying the characteristics of individual electrodes? This type of substrate is highly porous, how suitable is this for screen printing?
8. Why was 1M Na2SO4 chosen as the electrolyte?
9. The decision of the printing mode (SP-80) in forming the final supercapacitor is not entirely clear. The authors indicate that SP-80 has the highest capacitance (lines 235-236) at a current density of 0.3 A/g, but Figure 3d shows that the maximum capacitance is possessed by the SP-100 sample. At the same time, Figure S3 indicates that sample SP-80 has the longest discharge time, which should mean that it has the highest capacitance among the samples studied. This discrepancy should be resolved.
10. A typo in the ordinate axis caption in Figure 5i should be corrected.
11. It is necessary to clarify how the authors ensure the limitation of the interaction area between the electrolyte and the area of the nickel foam that is not covered with active material, because due to the capillary effect the liquid can rise on the substrate surface, wetting also the clean area of the nickel foam, which can also contribute to the overall capacitance.
Authors should check all the text and figures for typos.
Reviewer 2 Report
The study reports solvothermal synthesis of V3O7 nanopowder from V2O5, preparation and study of the inks based on the obtained V3O7 powder, and their application in screen printing of supercapacitors. Research was carried out on high methodological and instrumental level. Synthesis product has been meticulously studied and identified, using such techniques as SEM, TEM, SAED, XRD, XPS. Results from these techniques clearly show both microstructure (elongated nanorods) and composition (orthorhombic V3O7*H2O) of the product. The authors further carried out extensive research on rheological properties of inks based on V3O7, studying parameters important for practical application of said inks in screen printing of V3O7 coatings. Then, several oxide coatings were printed on various substrates, including a flexible PET substrate. Authors demonstrated good properties of obtained supercapacitors and that their properties practically are not affected by bending the sample.
All in all, I recommend the manuscript for publication in Nanomaterials due to its novelty, good obtained results and conclusions backed by very adequate analysis techniques. However, before publication, a few minor things should be addressed:
1. In Introduction, a couple sentences should be added detailing why solvothermal route was used.
2. In section 2.2 authors say that during synthesis "0.05 g V3O7 (served as template)" were added to reaction mixture, but no information on this V3O7 is given in section 2.1. This should be rectified.
3. Further in the manuscript, i.e. during V3O7 mechanism formation (lines 167-171), role of V3O7 template is not discussed at all. What function did it perform? Was it facilitating nanorod formation, or did it serve as seed for formation of specifically V3O7 and not, for example, VO2?
4. In Fig. 1b nanorod width is 124 nm, while in text authors, referring to this Figure, say that width is roughly 200 nm. That is roughly 100 nm, not 200 nm. Perhaps this sentence should be rewritten, indicating that while based on SEM nanorods are ~200 nm wide, TEM results show that they can also be close to 100 nm in width.
5. Careful examination of diffraction pattern in Fig. 7f reveals two low intensity peaks, which do not seem to have counterparts on PDF #85-2401. These peaks are at ~15 deg. (there's one peak in PDF card and two in experimental result) and at ~25-25.5 deg. Can these peaks be attributed to a small admixture of not completely reduced V2O5, or maybe to a V3O7 template that underwent partial reduction to VO2? Can V4+ to V5+ ratios from XPS provide any information on presence of such phases?
6. Manuscript is written very well in a good, easily understandable English, but here and there a few typos can be encountered. For example, in lines 161 and 204-205 authors wrote "future" when they clearly meant "futher". Such typos should be corrected, if possible.
Manuscript is written very well in a good, easily understandable English, but here and there a few typos can be encountered. For example, in lines 161 and 204-205 authors wrote "future" when they clearly meant "futher". Such typos should be corrected, if possible.
Reviewer 3 Report
Through the papere there is something wrong with the reference to a number of the figure, please check (line 133, 207, 212, 231, 263 please correct)
Table 1 is missing.
Round 2
Reviewer 1 Report
I thank the authors for answering the questions and revising the article according to the comments!